# K-ONE Playground: Reconfigurable Clusters for a Cloud-Native Testbed

**Jun-Sik Shin** [1] and **JongWon Kim** [2,*]

[1] School of Electrical Engineering and Computer Science, Gwangju Institute of Science and Technology (GIST), Gwangju 61005, Korea; jsshin@nm.gist.ac.kr

[2] AI Graduate School, Gwangju Institute of Science and Technology (GIST), Gwangju 61005, Korea

* Correspondence: jongwon@nm.gist.ac.kr; Tel.: +82-62-715-2219

**Abstract:** Cloud-native computing with edge clouds is dominating the current computing paradigm. To prepare a flexible testbed for this paradigm, the build-out of K-ONE Playground started in 2015 based on the concept of SmartXPlayground. K-ONE Playground targets a multi-site edge cloud testbed based on the concept of composable playground that can flexibly compose physical, virtual, container resources from a resource pool to user-defined infrastructure. SmartX Playground should properly handle demanding requirements for a composable playground. In this paper, we propose a unique design of reconfigurable clusters, which can provide physical and virtual resources ready for cloud-native DevOps services. We also describe a detailed implementation of the reconfigurable cluster for the real-world infrastructure of K-ONE Playground. Finally, we verify its feasibility with operations and practical examples of cloud-native service development.

**Keywords:** composable testbed; reconfigurable clusters; DevOps automation; template-based provisioning; multi-layered visibility

## 1. Introduction

Emerging cloud-native computing is an approach to build and orchestrate application services as interconnected containers over clustered compute resources [1]. Edge clouds are typically defined as small-sized and distributed clouds that can accommodate application services close to end devices [2]. Thanks to their strong advantages, applying cloud-native computing to edge clouds, known as cloud-native edge clouds [3,4], is becoming a popular option for operating diversified DevOps [5] services.

To verify their ideas about emerging cloud-native computing trends, software developers demand testbeds. By building a playground (i.e., testbed) infrastructure with distributed clusters, an ideal playground can provide distributed physical servers and virtual machines ready for cloud-native DevOps services. To build such a playground, we leverage the concept behind SmartX Playgrounds, which attempts a systematic approach to operate SDI (software-defined infrastructure)-oriented cloud testbeds in an automated way. Beyond building and operating stationary multi-site clouds, we are refining SmartX Playgrounds for composable playgrounds that can conceptually compose interconnected physical servers, virtual machines, and containers from distributed clusters, to support diversified DevOps services up to cloud-native computing flexibly. To be a composable playground, SmartX Playgrounds should properly cover demanding functional requirements. Even though public clouds could partly handle the requirements with extensive services, SmartX Playgrounds have different requirements because of resource limitations and target service domains. Besides, related work on cloud/edge cloud testbeds with successful use cases [6–9] is limited to specialized service domains.

For this reason, we propose reconfigurable clusters for composable K-ONE Playground as a multi-site cloud-native testbed. A reconfigurable cluster is a clustered resource pool whose collection of physical servers and virtual machines can be easily reconfigured for multiple tenants (i.e., operators and developers) using DevOps automation tools. We summarize the main contribution points of this paper as follows:

- We list the functional requirements of SmartX Playgrounds for composable playground. These requirements are clearly explained to provide a clear mapping with design choices for reconfigurable clusters effectively.
- To handle the functional requirements for a composable playground properly, we propose a unique design of reconfigurable clusters. We also provide a practical implementation of reconfigurable clusters for a real-world infrastructure.
- We describe how we operate reconfigurable clusters to provide different combinations of physical servers and virtual machines to multiple tenants easily. In addition, we verify the feasibility with practical use cases, which respectively demand different playground topologies for developing cloud-native DevOps services.

The rest of this paper is organized as follows. In Section 2, we provide the concept of SmartX Playground and list the requirements for a composable playground. In Section 3, we provide the design of reconfigurable clusters in terms of the overall concept, as well as the components and describe real-world infrastructure we target for implementation. In Section 4, we provide an implementation of reconfigurable clusters, which is followed by operations and utilization examples to show the feasibility of reconfigurable clusters in Section 5. We briefly summarize related work in Section 6, and finally, we conclude the paper.

## 2. SmartX Playground with Reconfigurable Clusters

In this section, we describe the basic concept of SmartX Playgrounds with the requirements for a composable playground.

### 2.1. SmartX Playground: Concept

SmartX stands for our research goal of providing flexible/agile user-defined services and infrastructure by leveraging SDI-oriented open source software and hardware. We have been developing and operating SmartX Playgrounds since 2009, as a part of the SmartX efforts, to align with rapidly emerging SDI-oriented services such as OpenFlow-based SDN, NFV, cloud, and cloud-native computing. As a result, we are operating SmartX Playgrounds such as OF@TEIN/KOREN Playground [10,11] and SmartX AI Cluster [12].

In our definition, SmartX Playgrounds [13,14] are miniaturized and customizable testbeds that can allow both operators and developers to develop and verify DevOps services with freely manipulable testbed resources. SmartX Playgrounds allow developers to pick/use resources and tools freely to enjoy service developments, comparing to workplace-like testbeds for development works, which provide fixed resources and tools with limited authority.

Figure 1 depicts the concept of SmartX Playgrounds. To build, operate, and utilize a multi-site playground with distributed boxes and clusters automatically, we have SmartX Playground Tower, which provides a logical space in a centralized location by following the concept of a monitor and control tower. SmartX Playground Tower can systematically cover various functional requirements for operating multi-site playgrounds by employing SmartX Automation Centers. SmartX Automation Centers are a software collection of Provisioning, Visibility, Orchestration, Intelligence, and Security centers that provide useful features to operators for intelligent operations of playgrounds. Provisioning center automates complicated procedures of the remote installation and configuration of distributed clusters. Visibility center provides multi-layered playground visibility that covers physical resources, virtual resources, and networking flow with a unified/interactive visualization support.

Orchestration center provides abstracted features of playground management by combining features of SmartX Automation Centers. Intelligence center manages high-performance clusters and schedules analytic tasks to support developing compute-intensive services such as deep learning and big data. Security center detects and identifies any suspicious activities by analyzing visibility data collected from distributed clusters.

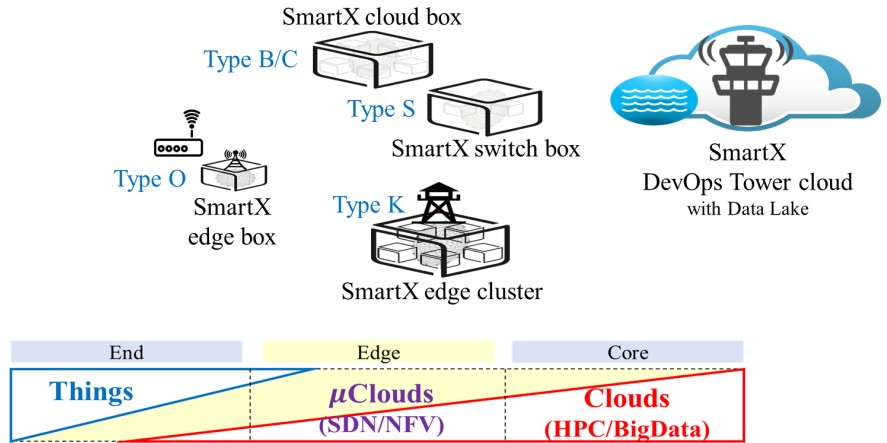

**Figure 1.** Concept of open source-leveraged SmartX Playground.

With the popularity of cloud and cloud-native computing, SmartX Playgrounds can be constructed with distributed clusters to support diversified DevOps services. In addition to SmartX Playground Tower, we introduce Post as a head entity of a cluster. Post closely monitors and controls application services, as well as resources in the cluster, like a security post at a higher location. In terms of operation, operators can utilize Post servers for intelligent operations, for example offloading operation workloads and autonomous cluster operations. On the other hand, Post servers can be useful for controller software of typical cloud-native DevOps services, which is sometimes called by different names such as coordinators, orchestrators, and masters. Therefore, from the experiences of operations, we define DevOps Post that allow multiple tenants to share the versatile Post servers concurrently. Clusters in playground sites also contain Cube that is implemented as multiple homogeneous SmartX boxes. A SmartX box is a hyper-converged box-style resource that collectively provides computing, storage, and networking resources as a single box-style entity. From Cube servers, SmartX Playgrounds can provide these integrated resources for diversified DevOps services.

In addition to these entities, we also divide SmartX Playgrounds and services into three main abstractions, which are box, function, and inter-connect. A box represents a physical/virtual resource entity that contains compute resources, networking resources, and storage resources. A function represents a software entity that can provide a specific feature of services. An inter-connect represents a physical/virtual path/link between two arbitrary entities. With the abstraction, we can logically simplify the operations of DevOps services. For instance, a procedure of deploying a typical three-tier service follows these steps: preparing boxes with inter-connects, putting web, App, DB functions in these boxes, and configuring inter-connects between these functions.

### 2.2. SmartX Playground: Requirements for a Composable Playground

As we mentioned in Section 1, our refinement of SmartX Playgrounds is based on the concept of the p (physical) + v (virtual) + c (container) Composable Playground. The p + v + c composable playground can view distributed clusters as a composable resource pool. From the pool, operators can flexibly compose physical/virtual/container boxes, functions, and their inter-connects to construct user-defined infrastructure for cloud-native DevOps services instantly. To follow the concept of a composable playground, SmartX Playground should properly handle the demanding requirements described as follows.

- R1 (Requirement 1). Providing the p+v+c harmonization testbed: The cloud-native computing paradigm introduces a container layer above physical/virtual layers. With the trend, typical cloud-based infrastructure and diversified DevOps services are adopting combinations of physical (p), virtual (v), and container (c) boxes. To simplify the complicated view, we introduce the concept of p + v + c harmonization, as shown in Figure 2. A composable playground should flexibly compose inter-connected boxes and functions across the layers, described as the dashed upward arrows in Figure 2, to support diversified services properly. Furthermore, a composable playground should clearly define actual forms of physical, virtual, and container boxes, as well as software tools for creating these boxes from distributed clusters.

- R2 (Requirement 2). Supporting multiple networks for flexible and reliable composition: For flexible and reliable composition from distributed clusters, a composable playground should support multiple networking connectivity within, as well as between clusters, while properly limiting the access of developers. Typical DevOps services can generate control traffic in addition to service data traffic for efficiency, reliability, and security. A composable playground should provide multiple networks for tenants to match this traffic to playground networks effectively. Meanwhile, heavy workloads and any incidents incurred from multiple tenants, especially in networks, and over shared clusters can result in failures of operations, as well as incorrect results of DevOps services. Thus, for a composable playground, SmartX Playground should separate operational traffic from service traffic.

- R3 (Requirement 3). Automating reconfiguration for a multi-site playground topology: A composable playground should automate the complicated procedures of playground topology reconfiguration for efficient operations. For playground topology reconfiguration, operators should understand the overall playground topology and properly compose unused boxes to satisfy the requirements of DevOps services. In a multi-site playground with distributed clusters, the reconfiguration can be more difficult due to the complicated topology and geographical separation. Thus, reconfiguring the playground topology without automation features can incur time waste and even human errors, which can disturb reliable operations and services. Thus, SmartX Playground should have automated features with DevOps tools to help operators at Tower servers easily grasp visibility and compose boxes from distributed clusters.

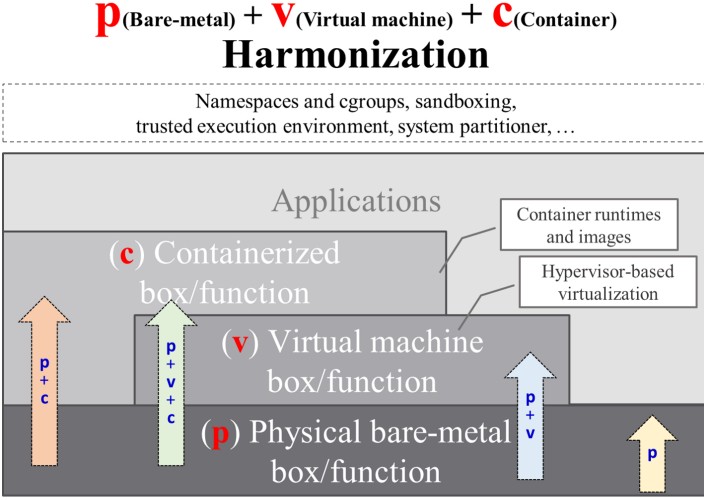

**Figure 2.** The concept of p (physical) + v (virtual) + c (container) harmonization.

## 3. Design of Reconfigurable Clusters

In this section, we provide the design of reconfigurable clusters. In addition, we describe the real-world infrastructure of K-ONE Playground where we implement reconfigurable clusters.

### 3.1. Overall Design of K-ONE Playground

To cover the requirements of a composable playground effectively, we propose reconfigurable clusters that are ready to provide partial resources in the forms of physical boxes and virtual boxes from distributed clusters. The concept of a composable playground and all of its requirements are difficult to accomplish in one sweep, so reconfigurable clusters focus on the easy composition of physical and virtual boxes for the desired topology by tenants. Figure 3 depicts the conceptual design of reconfigurable clusters. We designed reconfigurable clusters with components, such as preparing clusters for creating physical/virtual boxes, networking plane separation, and SmartX Automation Centers, to deal with the respective requirements properly.

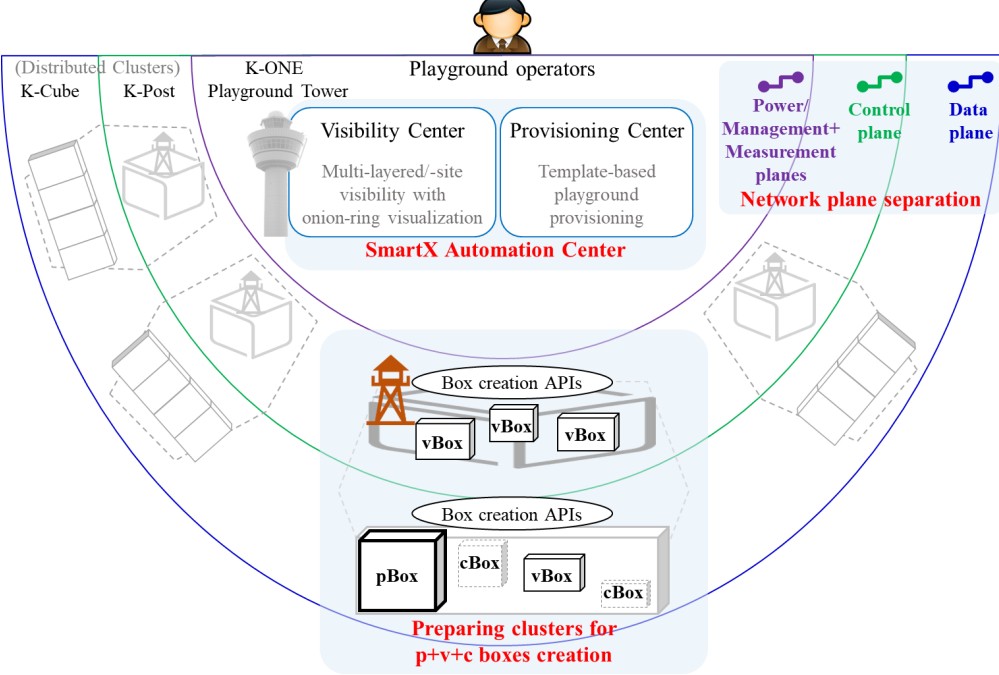

**Figure 3.** The overall design of reconfigurable clusters.

To handle R1, we carefully designed actual shapes of physical/virtual boxes and how to create these boxes from Cube servers, after discussing available options. In addition, we could configure a cloud over distributed Post servers to compose dedicated virtual boxes for multiple tenants to place controller software of cloud-native DevOps services easily.

We employed the concept of networking plane separation to satisfy R2. By categorizing networking traffic of typical DevOps services, we defined four networking planes, power, management, control, and data. As depicted in Figure 3, the entities and boxes on each half-circle should utilize the planes belonging for networking. This separation can limit accessible areas of tenants, so operators can ensure reliable networking for composing boxes. Furthermore, tenants can easily deploy typical cloud-native DevOps services by intuitively mapping control/data traffic to the separated planes.

For R3, reconfigurable clusters can utilize SmartX Automation Centers, especially Provisioning and Visibility centers, to reconfigure the playground topology automatically. For automation, we leveraged our DevOps tools such as the Distributed Secure Provisioning (DsP) Tool for Provisioning center and the SmartX MultiView Visibility Framework (MVF) for Visibility center. By employing these tools, Visibility center can provide a clear view of distributed clusters to find available boxes. Provisioning center can automate the procedures of software configurations to compose physical/virtual boxes remotely. Therefore, we could then easily identify used/unused resources of reconfigurable clusters and properly compose physical/virtual boxes from the unused slots.

*3.2. Components' Design for K-ONE Playground*

3.2.1. Reconfigurable Servers for Physical and Virtual Boxes

With the maturity of cloud/cloud-native computing, there are various candidates for physical, virtual, and container boxes. Thus, we designed physical and virtual boxes suitable for reconfigurable clusters by carefully selecting from available options, listed in Table 1.

**Table 1.** Available options for creating physical/virtual/container boxes.

|  | Options | Management Tools |
|---|---|---|
| **Container box** | Machine container | Machine container runtime (LXD) |
|  | Application container | Application container runtime (Docker) |
| **Virtual box** | Virtual machine | Virtual hypervisor (KVM, Xen) |
| **Physical box** | Bare-Metal Server | - |
|  | Partitioned cell | Hardware/system partitioner (ACRN Hypervisor, Linux Jailhouse) |

For physical boxes, we can consider two options. One is to provide bare-metal servers to tenants directly, and another is to provide cells that are divided by hardware (system) partitioners. Hardware partitioners can separate a physical server into multiple cells and explicitly allocate hardware components, such as CPU cores and networking ports, to these cells. With the multiple cells, a single physical server can accommodate multiple operating systems like virtual machines, but with strong hardware-level isolation. However, hardware partitioners such as ACRN Hypervisor [15] and Linux Jailhouse [16] are currently in the development phases, and they are not fully compatible with commodity servers and DevOps automation tools. Therefore, adopting hardware partitioners is not yet appropriate, so we allowed tenants to utilize whole bare-metal servers as physical boxes.

To compose versatile virtual boxes for diversified DevOps services, we can utilize virtual hypervisors such as KVM [17] and Xen [18], which have become highly mature along with the rapid growth of cloud computing. We selected the KVM (Kernel-based virtual machine) virtual hypervisor from the options due to its popularity and compatibility with various open source DevOps services.

As container boxes, system containers such as Linux container daemon (LXD) [19] can be suitable since system containers are designed to provide similar experiences of lightweight virtual machines, but using Linux containers. On the contrary, application containers such as Docker [20] are designed for containing application functions, not operating systems. In addition, isolating and limiting their resource usages are not as strict as for system containers. During playground operations, we served various tenants who developed cloud-native DevOps services, and they demanded to customize cloud-native clusters rather than directly taking system containers. Thus, instead of providing system containers and the runtime, we could provide physical/virtual boxes to tenants and let them configure customized cloud-native clusters by themselves.

In summary, the results of composing physical and virtual boxes can be bare-metal servers and virtual machines. The created boxes should be at least configured with Linux operating systems to allow tenants to utilize them instantly. Based on SmartX Playgrounds, we can logically simplify the procedures of creating boxes, as a simple task putting Linux operating system functions into empty boxes.

Notice that we are recognizing other options for boxes that are not mentioned in the list, such as Unikernel [21], Kata Container [22], gVisor [23], and Singularity [24]. These options are designed for special domains, and supporting all the variations can be difficult in terms of Playground operations.

Like the case of system containers, tenants can freely configure other options in the acquired physical/virtual boxes.

### 3.2.2. Networking Plane Separation

We designed four networking planes by categorizing heterogeneous traffics based on their characteristics: a power plane and a management + measurement plane for operators; a control plane and a data plane for tenants. Traffic types and use cases of the respective planes are described as follows:

- The P (Power) plane allows operators to maintain the hardware status of physical servers remotely, such as power status, temperature, event logs, and remote console access through the IPMI (Intelligent Platform Management Interface)-based remote management access [25]. The P plane can guarantee constant monitoring and control of physical servers even if other networking planes are unavailable due to, for example, operating system failures or shutdown.
- The M (Management + Measurement) plane is used for operating system management of physical boxes and virtual boxes. The M plane can accommodate SSH traffic to remote boxes for debugging and troubleshooting. Besides, SmartX Automation Centers in centralized Tower servers can use this plane to install and configure software packages for remote boxes. In addition, visibility data measured from boxes can be transferred to Visibility center through this plane.
- The C (Control) plane is mainly used for developers' control traffic generated from SDI-oriented services. For example, SDN controllers and switches exchange control and monitoring messages with each other and internal components in cloud/cloud-native computing exchange control messages with each other.
- The D (Data) plane can accommodate any service-level traffic of user experiments such as videos and voice data. With the D plane Playground, tenants can handle control traffic and data traffic, respectively. Thus, the developers can acquire relatively accurate results regarding their experiments.

### 3.2.3. SmartX Provisioning and Visibility Centers

The DsP tool is a template-based provisioning tool for Provisioning center that allows us to install and configure a customized playground topology automatically [26]. The DsP tool should cover bare-metal installation, virtual machine creation, and software packages' installation, for automatically composing physical/virtual boxes. For the DsP tool, we could leverage existing open source installation/configuration tools as much as possible. This choice was expected to let us stick to the DevOps-based automation paradigm while allowing us to enable various lightweight customizations of the DsP tool. We refined the DsP tool to satisfy changing requirements through operations of SmartX Playgrounds, which will be specifically described in Section 5. Using the DsP tool, Provisioning center can automatically provision (i.e., install and configure) physical boxes and virtual boxes from remote clusters.

SmartX MVF can provide multi-layered visibility of the overall playground with an onion-ring visualization dashboard [10,14,27–29]. The onion-ring dashboard of SmartX MVF was originally adapted to OF@TEIN Playground, which had a fixed playground topology of distributed physical servers with pre-installed multi-region clouds for SDN-cloud DevOps services [11,30]. Unlike OF@TEIN Playground, the reconfigurable clusters consider the multi-site edge cloud model and can frequently change the playground topology. Thus, to visualize the reconfigurable clusters intuitively, we should implement another version of the onion-ring visualization dashboard, as shown in Figure 4. The dashboard can initially depict distributed clusters as a big mass of available resource boxes, which is the grey-colored areas. When we compose boxes for tenants, the boxes can be visualized as partial rings divided by lines on the resource mass. Therefore, we can easily identify unused resources, as well as allocated resources from Visibility center.

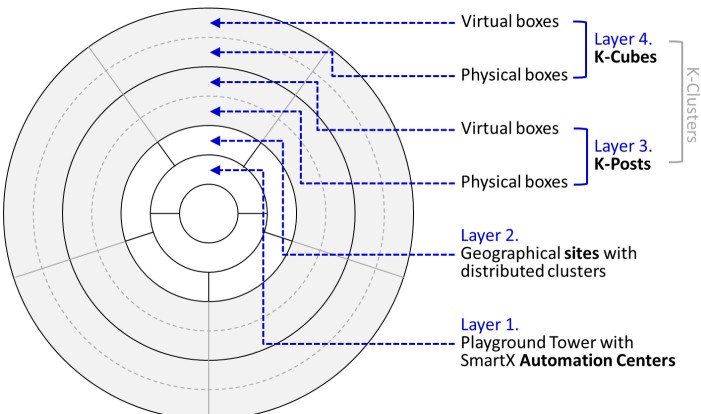

**Figure 4.** The design of the onion-ring dashboard for reconfigurable clusters.

### 3.3. Resource Infrastructure with Distributed Clusters

To cope with emerging cloud-native edge-computing, we have constructed a playground infrastructure with distributed clusters since 2015. Figure 5 describes the configuration of the playground infrastructure in detail.

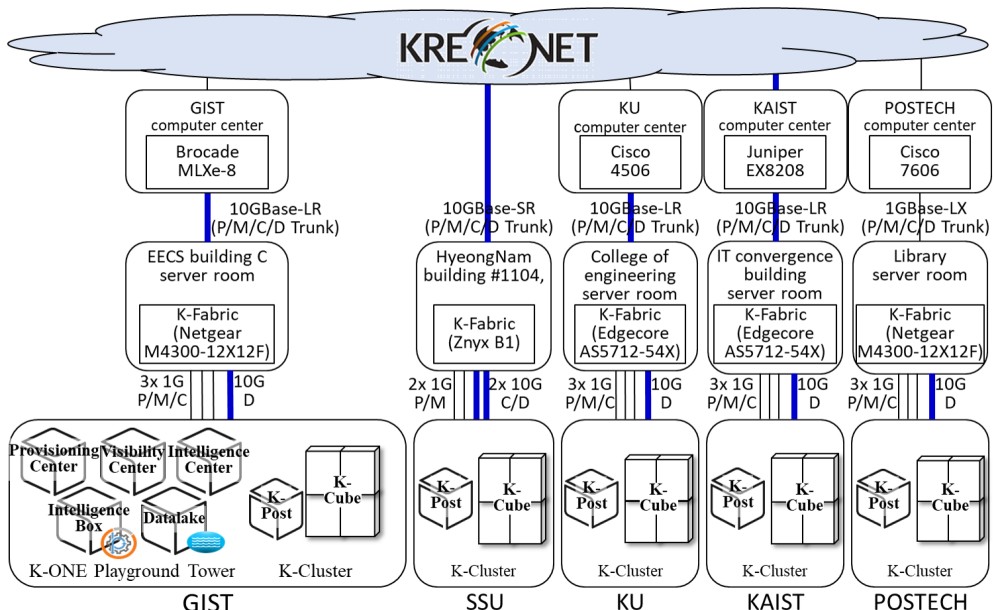

**Figure 5.** Resource infrastructure for K-ONE Playground. P, Power; M, Management + Measurement; C, Control; D, Data.

As a first step, we designed a small-sized cluster referred to as K-Cluster. K-Cluster consists of logical components such as K-Post, K-Cube, and K-Fabric. K-Cube can correspond to Cube of SmartX Playgrounds where tenants can utilize cloud-native DevOps services. K-Post follows the concept of Post in SmartX Playgrounds, which can accommodate operation tools, as well as controllers for managing boxes and services. K-Fabric tightly connects these physical servers with high-speed networking to form a clustered resource pool. We implemented K-Cluster with heterogeneous servers to suit the small-scale clusters: a 2U (rack unit)-sized server for K-Post, a NOS-supported switch for K-Fabric, and four mini-size servers for K-Cube.

We constructed resource infrastructure by deploying hardware implementations of K-Cluster at five universities in South Korea (i.e., GIST (Gwangju Institute of Science and Technology), KU (Korea University), SSU (Soongsil University), POSTECH (Pohang University of Science and Technology),

and KAIST (Korea Advanced Institute of Science and Technology)). To inter-connect the distributed clusters, the clusters were physically attached to KREONET (Korea Research Environment Open NETwork), which can support 10Gbps-capable wide area networks over South Korea. To configure K-ONE Playground Tower, we installed additional physical servers at the GIST site, which were exclusively used for SmartX Automation Centers.

## 4. Implementation of Reconfigurable Clusters

In this section, we provide an implementation of the proposed reconfigurable clusters. The implementation was iteratively refined during playground operations based on the DevOps-based automation methodology. That is, we operated the developed components over the real-world infrastructure, debugged, and refined them to troubleshoot the operation issues.

### 4.1. Preparing Distributed Clusters to Compose Physical and Virtual Boxes Remotely

In Section 3.2.1, we define shapes and local tools for creating physical and virtual boxes from physical servers. To enable clusters to be reconfigurable for physical and virtual boxes, we should implement RESTful APIs (Application Programming Interfaces) of local tools that are exposed to Post servers and Tower servers to create boxes remotely. By implementing the local tools and the RESTful APIs, clusters are reconfigurable in terms of creating physical and virtual boxes.

From K-Cube servers, SmartX Playgrounds should compose physical and virtual boxes. To implement the local tools for virtual boxes, we installed KVM, Virsh (with Libvirt), and clouds in physical boxes from K-Cube servers. However, for physical boxes, empty physical servers cannot solely install Linux operating systems themselves without external supports such as manual installation and bare-metal installation tools. We leveraged customized open source DevOps automation tools such as bare-metal installation tools and configuration management tools, rather than developing these tools and RESTful APIs from scratch. Implementations of these tools have been refined during operations, so we describe these tools in Section 4.3.

Post servers in SmartX Playground can be utilized for multiple tenants, as we describe in Section 2. Direct sharing of Post servers can incur an unstable and vulnerable status due to failures caused by tenants, such as software version conflicts and human errors. Instead, creating virtual boxes and inter-connects allows multiple tenants to acquire dedicated boxes from the shared Post servers. However, managing virtual boxes in distributed clusters involve cumbersome management tasks. To get around the issue effectively, we implemented a cloud cluster over distributed K-Post servers as shown in Figure 6 by leveraging OpenStack, which is one of the popular open source cloud operating systems [31].

Over distributed clusters in different geographical sites, a multi-region cloud can be a typical option for constructing a cloud infrastructure. In multi-region clouds, individual clouds are built on different sites, and identity services and RESTful APIs of the clouds are federated. Creating virtual boxes from multi-region clouds is more complicated than creating from a single cloud since we should manage virtual boxes, images, and virtual networks in the individual clouds. Besides, operating multi-region clouds is not suitable for reconfigurable clusters to simplify distributed Post servers logically as a pool of virtual boxes. Therefore, we implemented a single cloud with specially prepared L2 networks between the distributed K-Post servers. To accommodate diversified DevOps services, the cloud supports flat networks in addition to virtual local area network (VLAN)- and virtual extensible local area network (VXLAN) -based overlay virtual networks. Virtual boxes on the flat networks directly connect to physical boxes through L2 networks, so DevOps services can utilize virtual boxes in the same way as physical boxes.

Through the web dashboard of the cloud, we simply managed the life cycle of virtual boxes (i.e., creation, update, and deletion), as well as cumbersome tasks, for example virtual networking, virtual machine images, and tenants' accounts. Thus, we easily provided multi-site virtual boxes

from the K-Post servers, which helped tenants to implement cloud-native DevOps services in real-world infrastructure.

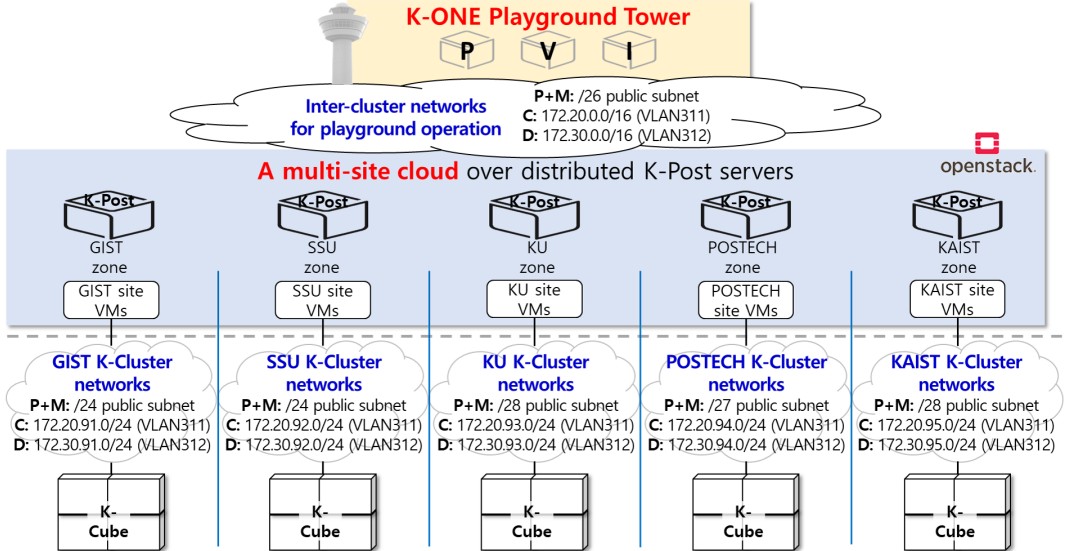

**Figure 6.** The configuration of a multi-site cloud over distributed K-Post servers.

## 4.2. Networking Plane Separation

There are many available approaches for networking plane separation. We employed a combination of physical and logical networking configurations. Notice that Figures 5 and 6 also depict the detailed implementations of networking plane separation.

To separate the networking planes physically inside a cluster, we equipped all physical servers with four networking ports: a 1 Gbps IPMI port for the P plane, two 1 Gbps ports for the M and C planes, and a 10 Gbps port for the D plane. These ports were connected to a K-Fabric switch. On the K-Fabric switch, we separated the planes by configuring different VLANs and IP subnets. In typical multi-site infrastructure, private networks of distributed clusters are isolated from each other, and therefore, servers at different sites cannot directly communicate through L2 networks. To reflect the typical configuration, we assigned different private IP subnets for the C and D planes to clusters. However, we configured public IP subnets to the P plane and M plane, to allow us to manage servers from centralized Tower servers remotely.

In addition to the intra-cluster networks, we additionally configured special L2 networks between the K-Post servers and K-ONE Playground Tower servers as we mention in Section 4.1. These L2 networks inter-connect the clusters through multiple networking planes and support to construct a cloud over the K-Post servers. For the inter-cluster networks, we configured an additional public subnet for the P + M planes. The inter-cluster C and D planes were configured as IP supersets of the intra-cluster C and D planes of all distributed clusters, due to the requirements of the OpenStack configuration.

Thanks to the networking plane separation of intra-/inter-cluster networking, we reliably operated the reconfigurable clusters. We as operators mainly utilized the P+M planes and inter-cluster C/D planes to operate various DevOps automation tools, as well as to reconfigure the playground topology. On the contrary, composed physical and virtual boxes are connected to the intra-cluster C and D planes. Therefore, operation tasks are not easily affected by traffics from tenants' DevOps services. Besides, tenants can intuitively map control and data traffic of DevOps services to the real-world infrastructure with C and D planes.

### 4.3. SmartX Provisioning and the Visibility Center

For Provisioning center, we implemented the DsP tool based on our experiences in developing and operating DevOps automation tools. In Figure 7, we depict the implemented software structure of the DsP tool with a flowchart describing the provisioning procedure. When we execute the DsP tool with a playground template, the DsP main module passes the template to the Playground Template Interpretation module. The playground template file describes the desired playground configuration. The Playground Template Interpretation module extracts the target list of distributed boxes with desired software collections from the given playground template file. The module also enriches the list of boxes with detailed box information (i.e., access credentials, network configuration) that is retrieved from Information Store module. The interpretation result is a list of boxes with software collections, as well as detailed box information, which specifically describes the desired configuration of the playground topology.

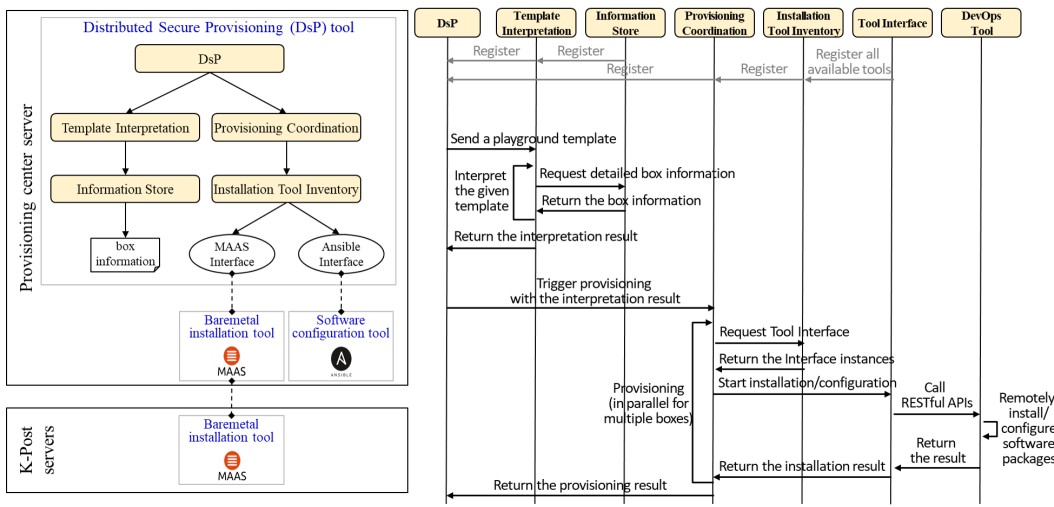

**Figure 7.** An implementation of the DsP tool (**left**: software structure; **right**: working procedure).

With the interpretation result, the DsP main module triggers the actual procedure of installation and configuration by invoking the Provisioning Coordination module. The Provisioning Coordination module reads the interpreted template and creates as many multiple processes as depicted in the template for concurrent provisioning of multiple boxes. These processes take the Installation Tool Interfaces matching the software collections from the Installation Tool Inventory module and invoke software installation tasks using the Interfaces one by one. The Installation Tool Interfaces are automatically registered to the Installation Tool Inventory module during the initialization process. The invoked Interface calls the APIs of the Installation Tools with the desired configuration of the target boxes. The Tool then automates the detailed tasks to install software on the target box. After finishing, the Coordination module reports provisioning results such as failures and the elapsed time on the display. In addition to the installation, the DsP tool also provides a feature to remove physical boxes by filling the storage disks of physical servers with zeros.

When it comes to Visibility center, we adjusted the onion-ring dashboard of SmartX MVF to suit intuitively visualizing reconfigurable clusters. Figure 8 depicts the overall playground topology visualized by the adjusted dashboard. The dashboard visualizes K-ONE Playground Tower servers for Provisioning and Visibility centers at the center. The geographical sites are placed on the next layer. Next, K-Post servers and K-Cube servers in distributed clusters are represented as grey-colored rings on the outer layers. Unlike the design, the dashboard visualizes virtual boxes on outer layers above the grey-colored rings, since nesting virtual boxes into the physical boxes complicates the structure of the onion-ring graph, preventing us from easily grasping the playground topology.

After reconfiguration, the onion-ring dashboard visualizes composed boxes as colored ring segments separated by colored lines. The boundary colors identify respective tenants who occupy the resources. The colors of the areas represent the status of the boxes: green for normal status, yellow for boxes that are powered off, and red for boxes that are not reachable through the P and M planes.

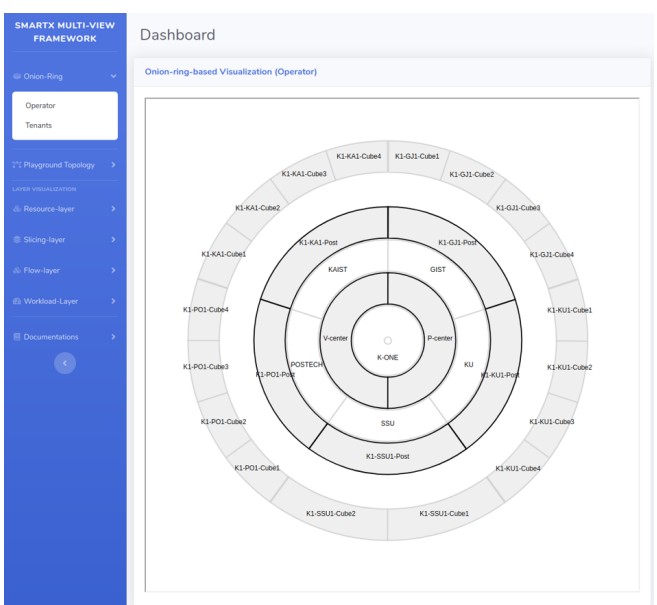

**Figure 8.** Implementation of the onion-ring visualization dashboard for K-ONE Playground.

## 5. K-ONE Playground: Operations and Utilization

In this section, we verify the feasibility of the proposed reconfigurable cluster by describing how to operate K-ONE Playground to compose physical and virtual boxes for tenants. We also depict practical use cases of K-ONE Playground that demand different playground topologies to develop cloud-native DevOps services.

### 5.1. Operations of K-ONE Playground

We have been operating K-ONE Playground based on the concept of composable playgrounds to provide user-defined infrastructure for multiple tenants since 2015. Over the period, we have supported tenants who usually want to utilize dedicated physical, virtual boxes in different combinations for developing their DevOps services. To support these tenants while satisfying the requirements, we operated the features of reconfigurable clusters to compose sets of physical and virtual boxes.

We describe the detailed reconfiguration steps of K-ONE Playground in Figure 9 Initially, Visibility center with SmartX MVF repeatedly collects visibility metrics from the distributed clusters and stores the data in the databases. The collected data are visualized on the onion-ring dashboard, so we easily understand the playground topology and status of distributed clusters.

The provisioning procedure is triggered by a request from Playground tenants. The tenants send descriptions of the desired box topologies to us, which list the number of boxes, their type (i.e., physical, virtual), and locations. From the onion-ring dashboard, we find available resources and translate the received request into a playground template that complies with the DsP tool's template format. To serve multiple tenants in the limited resources of K-ONE Playground, we follow the first-come first-served policy. Tenants should wait until distributed clusters become available to satisfy the requirements. As described in Figure 10, the YAML-based template format is simple, which lists target boxes with their required software packages. We call Provisioning center with the template to create boxes from distributed clusters. After provisioning, we confirm the result of the

provisioning by checking reports from the Provisioning center and the updated onion-ring dashboard of the Visibility center. We provide the access information of the created boxes to the tenants, so they can fully utilize the boxes.

To reconfigure the playground topology for the next tenants, the Provisioning center provides a feature of releasing created boxes. The release procedure should uninstall remaining software, in order to return these boxes to clean states. However, to clean up all software remaining after the previous tenants can be very complicated, since we should track all commands by tenants, analyze them, and clear the changes. Therefore, the Provisioning center does not cover releasing physical and virtual boxes at the fine-grained level. Instead, to release physical and virtual boxes in K-Cube servers, the Provisioning center utilizes the DsP tool to clean up physical disks. For the K-Post servers, we simply remove virtual boxes given for tenants using the APIs of the cloud.

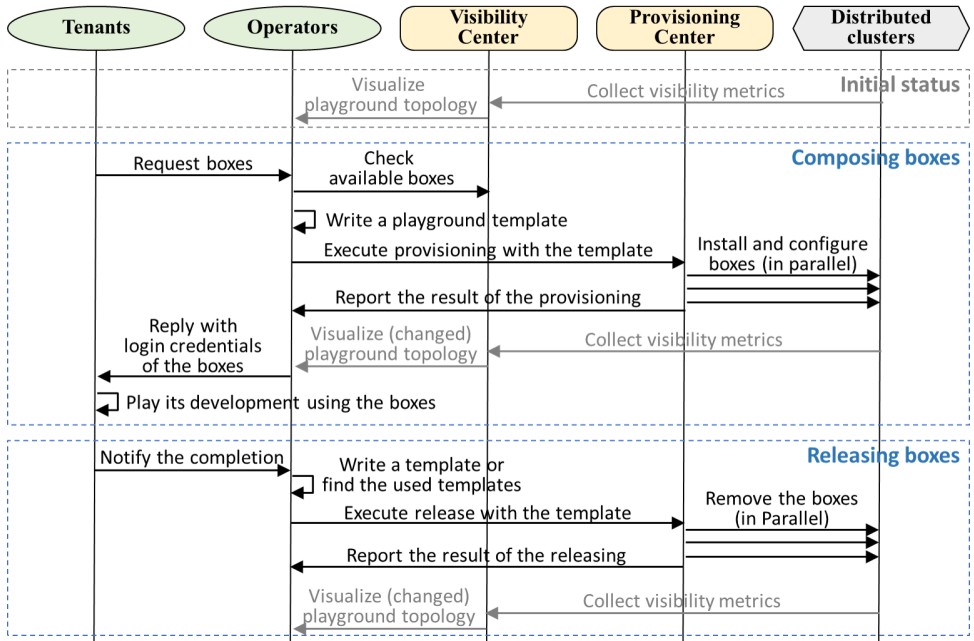

**Figure 9.** An operations the workflow of K-ONE Playground.

```
- tenant: <Tenant name>
  boxes:
   - name: <Box name>
     where: <A place for the box>
     type: <The type of the box>
     software:
      - name: <Software name>
        installer: <An Installer for the software>
        opt:
           <Detailed Options for the software>
```

**Figure 10.** Playground template format.

We refined the DsP tool to overcome operational issues based on the DevOps methodology. The first version of the DsP tool supported template-based automation of OpenStack-based cloud installation. For the tool, we defined three playground templates, which described the fixed topology of clouds over distributed servers. The DsP tool leverages open source DevOps automation tools such as Cobbler [32] for bare-metal installation and Chef [33] for cloud installation. However, we encountered operation issues such as slow adoption of emerging technology due to Chef's complexity and incompatibility problems between kernel versions and software packages.

To address the issues, we decided to leverage other open source tools, Canonical MAAS (metal as a service) [34] and DevStack [35]. MAAS is a bare-metal installation tool that can help in the easy operations of data center servers with handy features of server inventory management and operating system (OS) image management, through a GUI-based web dashboard. DevStack is a collection of shell scripts that allows developers to bring up an OpenStack development environment easily. When we were searching for a suitable tool for OpenStack installation, DevStack could easily deploy OpenStack along with emerging DevOps services such as SDN controllers and cloud storage. However, we again arrived at failures of cloud provisioning and operations since DevStack is not suitable for stable operations. DevStack does not strongly ensure successful OpenStack installation due to frequent updates of the scripts. We should repeat our installation tasks until clouds are correctly installed. For that reason, we implemented our shell script tool that followed the official manual for production-level OpenStack installation.

The latest version of the DsP tool supports two Interfaces for Canonical MAAS and Red Hat Ansible [36], as also described in Figure 7. We deployed the DevOps tools, MAAS region controller, and Ansible as system containers in the Provisioning center to prevent software conflicts between these tools. When it comes to MAAS, we deployed cluster controllers of MAAS to the respective K-Post servers as virtual boxes to distribute operation workloads from the centralized Tower servers. The region controller in the Provisioning center provides a web-based dashboard and RESTful interfaces, and the cluster controllers conduct actual installation tasks of physical boxes using IPMI and PXE booting. In addition to MAAS, we utilized Ansible to compose virtual boxes, as well as to install software packages by leveraging its versatile features. That is, composing and releasing virtual boxes were implemented as Ansible playbooks, which describe the detailed steps of software configurations for Ansible to automate provisioning procedure.

To verify the feasibility, we measured the elapsed time to reconfigure the playground topology with an example scenario. In this scenario, we assumed that two tenants demanded to acquire boxes with different playground topologies. The first tenant demanded four physical boxes from two sites and two virtual boxes inside the created physical boxes. Thus, we composed four physical boxes from two sites, installed two OpenStack-based clouds, and composed two virtual boxes from these clouds. After finishing the tenant's development, we released the physical boxes. Next, another tenant demanded two virtual boxes in two K-Post servers, so we composed and released these virtual boxes from the cloud over the K-Post servers.

To measure elapsed times of each step, we repeated the scenario using GIST and POSTECH clusters five times. The reconfiguration for the first tenant utilizes the DsP tool and the installation tools (i.e., MAAS region controller and Ansible), which work as LXD system containers in the Provisioning center. To compose physical boxes, the DsP tool invokes MAAS cluster controllers in GIST and POSTECH clusters, which work in the K-Post servers as OpenStack VMs having one virtual CPU and 2 GB memory. Then, the MAAS cluster controllers automatically install Linux OS (Ubuntu 18.04) and other software packages for the selected K-Cube servers. The K-Cube servers are equipped with one Intel Xeon-D CPU (2.2 GHz, 4-cores), 32 GB memory, and 512 GB SSD. The DsP tool invokes Ansible using the Ansible interface to install and configure two OpenStack clouds automatically. Next, two virtual boxes with two virtual cores and 4 GB memory are composed on the configured clouds. Finally, the DsP tool releases the composed boxes by cleaning the disks of the physical boxes. Lastly, the DsP tool removes the composed virtual boxes. For the second tenant, Ansible invoked by the DsP tool composes two virtual boxes with two virtual cores and 4 GB memory in GIST and POSTECH K-Post servers by calling RESTful APIs of the multi-site OpenStack cloud.

The results on average are represented in Table 2. From K-Cube servers, K-ONE Playground can compose multi-site physical boxes within 10 min and virtual boxes with OpenStack clouds around 16 min. Besides, composing multi-site virtual boxes in the K-Post servers only takes around 2 min. However, the release of physical boxes can be varied depending on the size and type of physical disks. Regardless of the number of boxes, we can approximately calculate the reconfiguration time

for a specific playground topology by simply adding up the times, since the Provisioning center composes multiple boxes in parallel. Besides, we concluded from the result that K-ONE Playground could reconfigure the distributed clusters to the desired playground topology in at most 30 min by executing a single command. Notice that we focused on verifying the feasibility of the proposed features, not performance, so reducing the configuration time was out of our focus in this paper.

**Table 2.** Time measurements of playground topology reconfiguration.

| Scenario | | Elapsed Time |
|---|---|---|
| **1-1. Composing multi-site physical boxes from K-Cube servers** | | 8 min 58 s |
| **1-2. Composing virtual boxes on cloud-enabled physical boxes** | 1-2-1. OpenStack control node | 14 min 33 s |
| | 1-2-2. OpenStack compute node | 5 min 40 s |
| | 1-2-3. Virtual boxes | 1 min 12 s |
| **1-3. Release physical boxes** | | 3 min 39 s |
| **2-1. Composing multi-site virtual boxes from the K-Post servers** | | 1 min 29 s |
| **2-2. Release the virtual boxes** | | 16 s |

*5.2. K-ONE Playground Utilization*

The main features of the reconfigurable clusters are to compose physical/virtual boxes suited for diversified cloud-native DevOps services easily, with intuitive visibility support. In this section, to present use cases of these features, we describe practical utilization examples of developing cloud-native DevOps services. Furthermore, we describe how the reconfigurable clusters could be visualized and easily customized to match the desired topology for the examples.

5.2.1. Utilization #1: Multi-Site Physical Boxes for Cloud-Native Dynamic Overcloud

In the first case, a tenant demands multi-site physical boxes for the development of the concept of dynamic overcloud that can provide service compatibility regardless of types and locations of underlay clouds, as depicted in Figure 11. For the compatibility, the overcloud tower can dynamically configure dynamic overcloud, which is an additional layer between a service layer and a cloud infrastructure layer, by deploying Kubernetes-based cloud-native clusters, visible fabric, and the connected data lake over hybrid multi-clouds such as the Amazon Web Services public cloud and OpenStack private clouds. The tenant demands a playground topology with six physical boxes from two geographical sites for two OpenStack private clouds and an additional virtual box for overcloud tower.

A playground template describes the desired playground topology: three physical boxes from the GIST cluster, three physical boxes from the POSTECH cluster, and a virtual box from a K-Post server. The Provisioning center with the DsP tool installs operating systems for the selected physical boxes and instantiates a virtual box in the GIST K-Post server. As a result, the onion-ring visualization shows the reconfigured playground topology as depicted in Figure 12. Using the multi-site physical boxes, the tenant could successfully develop DevOps automation tools for cloud-native dynamic overcloud on customized OpenStack-based clouds, which finally resulted in the research outcome [37].

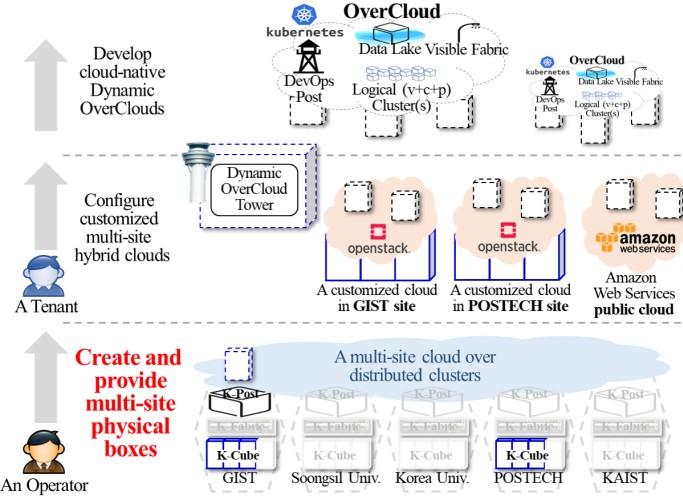

**Figure 11.** Multi-site physical boxes for cloud-native dynamic overcloud.

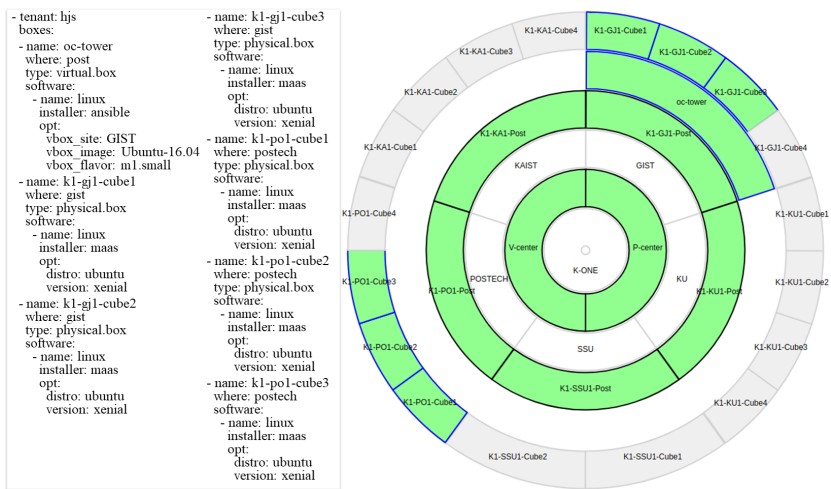

**Figure 12.** Playground reconfiguration for multi-site physical boxes: a template (**left**); the result (**right**).

### 5.2.2. Utilization #2: Multi-Site Virtual Boxes for Cloud-Native Service Mesh Service

Service mesh [38] is an additional layer between service functions and cloud-native infrastructures that can monitor and control service traffic without touching application codes. When it comes to a service mesh, monitor/control traffics in a service mesh can be visible to application functions, which may incur security problems. In the second practical use case, a tenant asks for multi-site virtual boxes to develop a protected coordination scheme for the service mesh that can separate the monitor/control traffic and data traffic to prevent such exposure. A playground topology for the desired testbed consists of four servers that are capable of L2-based networking, but placed in different sites, as depicted in Figure 13. A playground topology is written on the template in Figure 14, and four virtual boxes from K-Post servers are allocated to the tenant. The Provisioning center automatically creates two virtual boxes for the GIST site and another two for the Soongsil University and POSTECH sites, respectively. The reconfiguration results in the changed playground topology as visualized on the onion-ring dashboard shown in Figure 14. By utilizing the virtual boxes, the tenant configured a customized cloud-native cluster and could develop the DevOps service with the outcome found in [39].

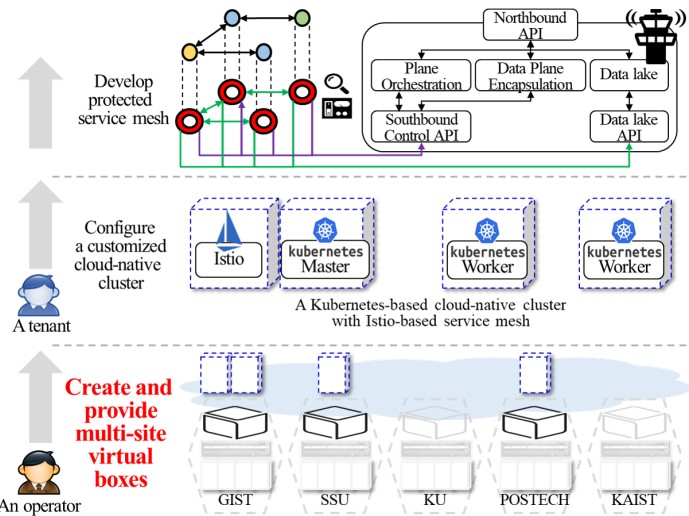

**Figure 13.** Multi-site virtual boxes for cloud-native multi-site service mesh.

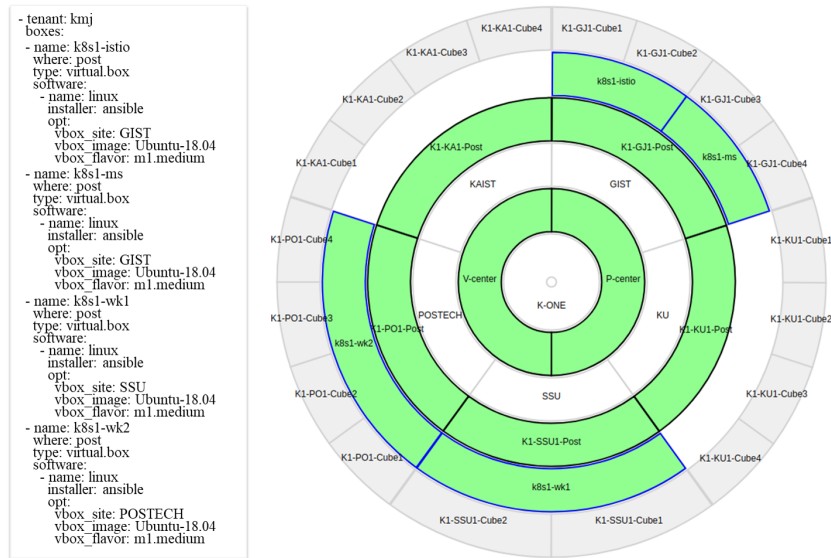

**Figure 14.** Playground reconfiguration for multi-site virtual boxes: a template (**left**); the result (**right**).

### 5.2.3. Utilization #3: Multi-Tenants Testbeds for Open Source Software Development

K-ONE Playground has been actively utilized to support various contribution efforts of SDI-oriented open source software projects. Figure 15 shows an example case of multi-tenant experiments that could be easily supported by utilizing the reconfigurable clusters. In the third use case, two tenants demand to utilize K-ONE Playground. A tenant focuses on developing a Docker Swarm plugin of the Open Baton framework software, and at the same time, another develops a monitoring system and anomaly detection service for M-CORD.

When it comes to the playground topology, two tenants respectively demand four physical boxes from two sites. In addition, one of the tenants wants a virtual box to place the Open Baton NFV orchestrator. Based on the requirements, we selected the locations of the physical/virtual boxes and wrote a playground template as shown in the left of Figure 16. The Provisioning center results in the reconfigured playground that can be intuitively visualized on the onion-ring dashboard shown in the right part of Figure 16. Among the green-colored elements, the blue-colored border belongs to the Open Baton developer, and the M-CORD developer can utilize the boxes with the red-colored border. We could properly provide collections of physical and virtual boxes to the tenants, and they could successfully achieve the research outcomes [40,41].

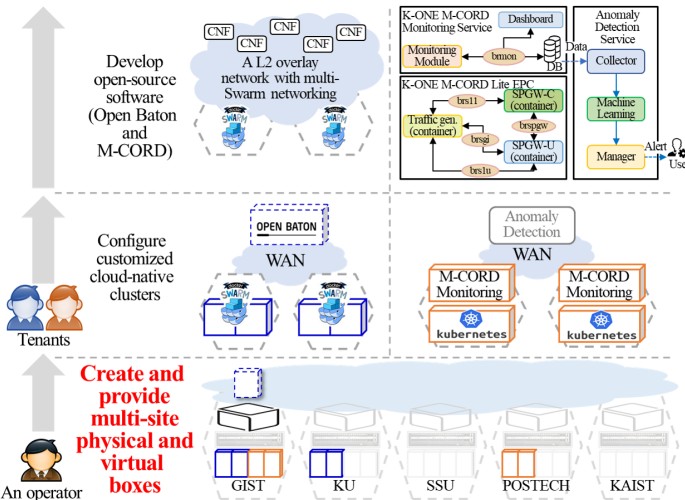

**Figure 15.** Multi-tenant support for SDI-oriented open source software development.

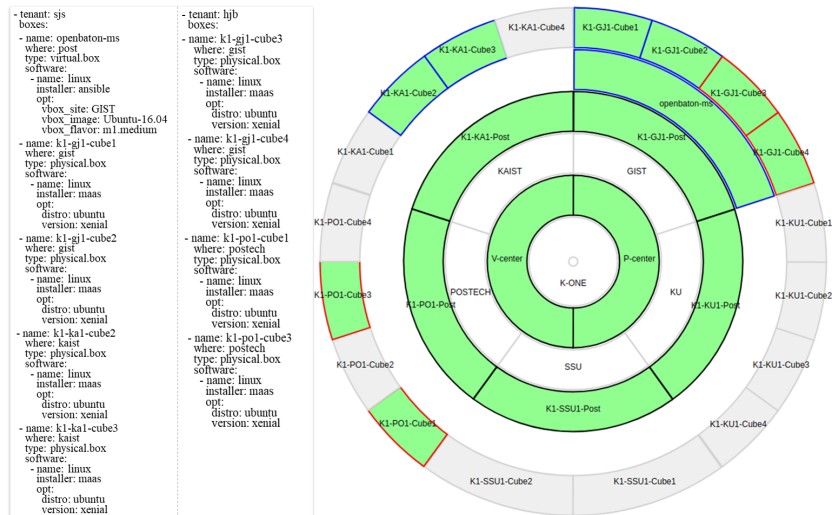

**Figure 16.** Playground reconfiguration for multi-tenant support: a template (**left**); the result (**right**).

## 6. Related Work

Public cloud services are currently supporting extensive features providing physical and virtual boxes to multiple tenants. However, the reconfigurable clusters have different requirements because of resource limitations and target services compared to public clouds. Public clouds typically possess abundant resource pools that are separately optimized for respective services. Thus, they can provide physical and virtual boxes to multiple tenants by simply selecting from the matching resource pools, which are optimized for each resource/service type. On the contrary, the reconfigurable clusters can customize distributed clusters on-demand to satisfy multiple tenants with a limited resource pool. Furthermore, DevOps services may demand freely manipulating wide levels of infrastructure software and even physical hardware devices, to cover both aspects of service development and operations. In public clouds, tenants entrust the operations of the underlying infrastructure to public cloud operators. Therefore, it can be difficult for tenants to acquire such levels of freedom from public cloud services, but K-ONE Playground can support these demands. Likewise, on-premise testbeds still have niche values and demands, so there are many research efforts to propose the build-out and operations of their testbeds for DevOps services.

Through various research, worldwide researchers have proposed cloud/edge cloud testbeds with successful examples of development use cases. The respective testbeds consider different infrastructure topologies, target service domains, and resources/tools for developers. Cumulus testbed [6] gives

developers a task dispatcher that can help deploy offloading services over physical resources and virtual resources of edge clouds. The Homecloud testbed [7] provides SDN/NFV-leveraged edge clouds with OpenStack-based NFV-cloud controllers and SDN controllers. PlanetIgnite testbed [8] provides virtual resources from distributed edge clouds to developers with a helper tool to deploy container-based services easily. These testbeds have in common that they can help tenants easily develop deployments/orchestration services for virtualized applications in different target domains. However, these testbeds are limited to providing software tools, controllers, or virtual resources that are not suited for supporting the customized configuration of cloud-native clusters. Thus, the approach of the testbeds is not well aligned to our requirements due to the differences.

Among the related work, the Smart Applications on Virtual Infrastructure (SAVI) testbed [9] has a similar approach to K-ONE Playground, in terms of the research domains and resources given to tenants. The SAVI testbed is a multi-tier heterogeneous cloud testbed that consists of federated core/edge clouds, WiFi access points, and IoT things. The testbed also includes a centralized software framework that can be similar to SmartX Automation Centers, and the framework utilizes an Ansible-based provisioning tool and visibility tool in order to configure, monitor, and visualize the overall testbed automatically. Furthermore, the authors also considered leveraging Kubernetes-based cloud-native computing for containerizing management services such as OpenStack components, OpenFlow controller, and other operational services. As shown above, the SAVI testbed and K-ONE Playground have many similar points, but our work focuses more on suggesting an easy and unique design for constructing a cloud-native-ready testbed where an operator and developers can enjoy the features of reconfigurable clusters. Furthermore, tenants of K-ONE Playground can intuitively map their topology of cloud-native clusters and even freely manipulate physical and virtual boxes.

## 7. Conclusions

In this paper, we proposed K-ONE Playground, which is a multi-site cloud-native-ready testbed with reconfigurable clusters. K-ONE Playground follows the concept of SmartX Playground, but it has limitations on addressing the additional requirements of reconfigurable clusters. Thus, we proposed a unique design with three essential elements that could make our distributed clusters reconfigurable: the definitions of physical/virtual boxes in reconfigurable clusters, networking plane separation for enhancing reliability, and SmartX Automation Centers with DevOps tools that could easily reconfigure the playground topology on-demand. We verified that the reconfigurable clusters could easily reconfigure the playground topology by describing actual use cases of K-ONE Playground. Even though the detailed implementation was customized for our testbed, K-ONE Playground may give an idea to researchers about how to build and operate a multi-site cloud-native-ready testbed easily.

**Author Contributions:** Conceptualization, J.-S.S. and J.K.; investigation, J.-S.S.; software, J.-S.S.; supervision, J.K.; validation, J.-S.S.; visualization, J.-S.S.; writing, original draft, J.-S.S.; writing, review and editing, J.-S.S. and J.K. All authors read and agreed to the published version of the manuscript.

**Funding:** This work was supported by the Institute of Information & communications Technology Planning & Evaluation (IITP) grant funded by the Korean government (MSIT) (No. 2015-0-00575, Global SDN/NFV Open Source Software Core Module/Function Development, and No. 2017-0-00421, Cyber Security Defense Cycle Mechanism for New Security Threats).

**Conflicts of Interest:** The authors declare no conflict of interest.

## Abbreviations

The following abbreviations are used in this manuscript:

| | |
|---|---|
| DevOps | Development and Operations |
| DHCP | Dynamic Host Configuration Protocol |
| DsP | Distributed Secure Provisioning |
| GUI | Graphical User Interface |

| IPMI | Intelligent Platform Management Interface |
| KREONET | Korea Research Environment Open NETwork |
| KVM | Kernel-based Virtual Machine |
| K-ONE | Korea-Open Networking Everywhere |
| LXD | LinuX container Daemon |
| LAN | Local Area Network |
| MAAS | Metal as a Service |
| M-CORD | Mobile-Central Office Re-architected as a Data Center |
| NFV | Network Function Virtualization |
| NOS | Network Operating Systems |
| OF@TEIN | OpenFlow at Trans-Eurasia Information Network |
| PXE | Pre-eXecution Environment |
| REST | Representational State Transfer |
| SDI | Software-Defined Infrastructure |
| SDN | Software-Defined Networking |
| MVF | MultiView Visibility Framework |
| SSH | Secure Shell |
| VLAN | Virtual Local Area Network |
| VXLAN | Virtual Extensible Local Area Network |
| VM | Virtual Machine |
| YAML | YAML Ain't Markup Language |

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
