# Peer review of "K-ONE Playground: Reconfigurable Clusters for a Cloud-Native Testbed"

_electronics, doi:10.3390/electronics9050844_

Round 1

Reviewer 1 Report

The paper proposes a unique design of reconfigurable clusters, which can provide physical and virtual resources ready for cloud-native DevOps services. They also describe a detailed implementation of the reconfigurable cluster for the real-world infrastructure of the K-ONE playground. In addition, they verify its feasibility with operations and practical examples of cloud-native service development.

The paper is well written. The methodologies used are appropriate and the results obtained with K-ONE are scientifically sound, bringing a significant contribution to the field. 

There are some issues that the authors need to address before the paper can be published:

1- The font size and colour used in the rectangle between the words Harmonization and Applications in Figure 2 is not appropriate. One can barely read what written inside.

2- Figures 3, 4, 9, 11, 13, and 15 also have a bad choice in terms of font size and colour in some parts of the figure (the font size is too small and the shade of grey used is too bright).

3- Figures 5, 6, 11, 13, and 15 need to be stretched because they are too small (maybe the fonts used could also be improved).

Author Response

I wrote down my responses to your valuable comments in the attached cover letter. Please see the attachment.

Reviewer 2 Report

This paper presents the design and implementation K-ONE playground, a reconfigurable cloud testbed.

The authors details the overall system design and implementation.  However, it is not clear what are the key innovations of this testbed.  Also, what are the motivations of the development of such testbed while many public cloud services are already widely available.  What are the use cases of such reconfigurable testbed?

In addition, there is no detailed system evaluation.  It is not clear what are the performance metrics, experiments, and evaluation results.  

Author Response

(The authors gave the same response as above.)

Round 2

Reviewer 2 Report

Most of my concerns have been addressed.